# The Multi-Target Action Mechanism for the Anti-Periodontitis Effect of *Astragali radix* Based on Bioinformatics Analysis and In Vitro Verification

**DOI:** 10.3390/nu17040627

**Published:** 2025-02-10

**Authors:** Ningli Li, Bowen Wang, Mingzhen Yang, Miaomiao Feng, Xiaoran Xu, Cory J. Xian, Tiejun Li, Yuankun Zhai

**Affiliations:** 1School of Stomatology, Henan University, Kaifeng 475004, China; liningli994@163.com (N.L.); yang-mingzhen@henu.edu.cn (M.Y.); fmm1660445237@126.com (M.F.); xu942478613@hotmail.com (X.X.); litiejun22@vip.sina.com (T.L.); 2Kaifeng Key Laboratory of Periodontal Tissue Engineering, Kaifeng 475004, China; wangbw512@163.com; 3UniSA Clinical and Health Sciences, University of South Australia, Adelaide, SA 5001, Australia; cory.xian@unisa.edu.au; 4Department of Oral Pathology, Peking University School and Hospital of Stomatology, Beijing 100081, China

**Keywords:** *Astragali radix*, periodontitis, network pharmacology, molecular docking, molecular dynamics simulation, kaempferol, in vitro experiments

## Abstract

**Background:** *Astragali radix* is a traditional Chinese medicine with potential therapeutic effects on periodontitis; however, its underlying mechanisms require further investigation. **Methods:** We employed network pharmacology, molecular docking, molecular dynamics simulations, and in vitro experiments to explore the potential actions and mechanisms of *Astragali radix* in treating periodontitis. **Results:** A total of 17 compounds (including the most prevalent one, Kaempferol) from *Astragali radix* and 464 corresponding targets were identified, from which five major active ingredients were selected based on the drug-active ingredient and periodontitis gene network. Protein–protein interaction (PPI) network analysis identified the top ten core potential targets, seven of which possess suitable crystal structures for molecular docking. These include interleukin-6 (IL6), tumor necrosis factor (TNF), AKT serine/threonine kinase 1 (AKT1), interleukin-1β (IL1β), prostaglandin G/H synthase-2 (PTGS2), matrix metalloproteinase-9 (MMP9), and caspase-3 (CASP3). Additionally, 58 Gene Ontology (GO) terms and 146 Kyoto Encyclopedia of Genes and Genomes (KEGG) pathways were identified. The five major active ingredients and seven core targets mentioned above were subjected to molecular docking analysis using Discovery Studio 2019 software. Molecular dynamic simulations confirmed a stable interaction between the CASP3 and the Kaempferol ligand system. In vitro experiments indicated that Kaempferol significantly inhibited lipopolysaccharide (LPS)-induced apoptosis in human periodontal ligament stem cells and reduced the expression levels of IL6, CASP3 and MMP9. **Conclusions:** This study systematically elucidates that the primary active ingredients derived from *Astragali radix* exert their pharmacological effects (including anti-inflammation and anti-apoptosis) primarily by interacting with multiple targets. These findings establish a promising foundation for the targeted application of *Astragali radix* in the treatment of periodontitis.

## 1. Introduction

Periodontitis, primarily characterized by the formation of periodontal pockets, alveolar bone resorption, and tooth loosening, is a highly prevalent inflammatory disease of the oral cavity [1]. Recent studies have highlighted the association between periodontitis and various systemic diseases, including rheumatoid arthritis, type 2 diabetes, atherosclerotic cardiovascular disease, and osteoporosis [2,3]. The current clinical management of periodontitis primarily emphasizes basic, pharmacological, and surgical treatments [4,5]. In these treatments, pharmacological interventions mainly serve as adjuncts to basic and surgical approaches, particularly in cases of acute periodontal infections, inaccessible infected sites, patients with systemic diseases, or individuals unable to maintain proper oral hygiene [5].

Traditional Chinese herbal medicine has garnered significant attention from researchers due to its long history of use, high safety profile, and low cost. It has recently emerged as a focal point in oral disease research, particularly for the treatment of periodontitis. *Astragali radix*, referred to as “Huangqi” in China, is the dried root of *Astragalus membranaceus* and is primarily sourced from mountainous regions characterized by relatively low precipitation [6]. Pharmacological studies have demonstrated that the biological activities of *Astragali radix* encompass immunomodulatory, anti-inflammatory, anti-diabetic, anti-viral, anti-tumor, cardioprotective, anti-hyperglycemic, anti-oxidant, and anti-aging effects, with minimal side effects [6]. *Astragali radix* is believed to be effective in treating a variety of common diseases, including diabetes, tumors, and autoimmune disorders [7]. *Astragali radix* is composed of a variety of chemical components, including Astragalus polysaccharide, saponins, and flavonoids (including the most prevalent one Kaempferol) [8].

Despite the recognition of *Astragali radix*’s clinical efficacy in recent years, there are limited reports on its pharmacodynamic basis and molecular mechanisms in the treatment of periodontitis. Additionally, systematic research involving animal and cellular studies to explore these mechanisms is lacking, which restricts the further development and application of *Astragali radix*. However, the integration of bioinformatics and network pharmacology may offer a practical approach to exploring these mechanisms of action of *Astragali radix* [9]. Network pharmacology is a research method based on systems biology theory, one that utilizes network mining data and integrates a large amount of information to explore drug targets and molecular mechanisms [10,11]. Therefore, we employed network pharmacology to elucidate the relationship between *Astragali radix* and periodontitis-related targets from a holistic and systematic perspective, aiming to reveal the possible molecular mechanisms by which *Astragali radix* exerts its therapeutic effect on periodontitis.

Molecular docking is a widely utilized method in computer-aided drug design that assesses the binding affinity between drug molecules and target proteins [12,13]. Molecular dynamics simulation is a technique that examines the time-resolved motions between receptors and small molecules. This technique is utilized to further investigate and evaluate the binding of ligands to receptor proteins, thus offering a more accurate understanding of their dynamics and interactions [14]. *Astragali radix* has been demonstrated to be effective in the treatment of periodontitis; however, its complex composition and specific mechanisms remain unclear. The novelty of the current study resides in the application of molecular docking, molecular dynamics techniques, and in vitro cell experiments to identify the primary active components of *Astragali radix* for periodontitis therapy and their associated disease targets, thereby offering a preliminary elucidation of its potential mechanisms. All these findings are anticipated to deepen the understanding of the molecular mechanisms underlying the role of *Astragali radix* during the treatment of periodontitis, thereby creating additional opportunities for its clinical application in near future for the prevention and management of periodontitis.

## 2. Methods and Materials

### 2.1. Identification of Bioactive Components and Corresponding Targets of Astragali radix

The traditional Chinese medicine system pharmacology database and analysis platform (TCMSP, https://tcmsp-e.com/, accessed on 18 October 2022) was built based on the framework of system pharmacology for herbal medicines [15]. Oral bioavailability (OB) represents the ratio of oral dose to unchanged drug reaching the systemic circulation and is one of the most important pharmacokinetic parameters [16], and drug-like properties (DL) is a qualitative concept used to estimate the drug-forming properties of a molecule [17]. Following the previously published data, the bioactive components and targets of *Astragali radix* were obtained according to the conditions of OB ≥ 30% and DL ≥ 0.18 [18].

### 2.2. Target Prediction of Astragali radix in the Treatment of Periodontitis

Target genes for periodontitis were searched through the GeneCards platform (https://www.genecards.org/, accessed on 25 October 2022) [19], OMIM platform (https://www.omim.org/, accessed on 25 October 2022) [20], and DrugBank platform (https://www.drugbank.ca/, accessed on 25 October 2022) [21]. Meanwhile, 2 datasets (GSE10334 and GSE16134) were obtained by searching periodontitis-related datasets in the GEO database (https://www.ncbi.nlm.nih.gov/geo/, accessed on 25 November 2022) using the keyword “periodontitis”, GSE10334 and GSE16134 [22,23,24]. Samples in both datasets were taken from gingival tissues, with samples in the experimental group with periodontitis, and samples in the control group from healthy gingival. These two datasets were unified using the GPL570 sequencing platform, and differential expression analysis was performed after batch correction to identify differentially expressed genes. Data correction was performed in Perl. The Limma package in R software (4.2.2) was used for differential gene expression analysis, and the ggplot2 package was used to plot volcano and heat maps. *p* < 0.05 and fold change (FC) > 2 was identified as a significant difference between the two groups [25,26]. GSE10334 contained 183 samples and GSE16134 contained 241 samples. All periodontitis-related targets were obtained and confirmed from the databases mentioned above. A crossover network of targets of active ingredients in *Astragali radix* and periodontitis disease-related targets was constructed using Venny 2.1.0 (https://bioinfogp.cnb.csic.es/tools/venny/, accessed on 25 November 2022) [27] to screen the intersection targets used for the following analyses.

### 2.3. Construction Networks

Cytoscape 3.9.0 software was used to construct the *Astragali radix* active components–corresponding targets network and the network of intersection targets between *Astragali radix* components and periodontitis [27]. The degree values of each node obtained in this network were analyzed topologically to identify the key active ingredients of *Astragali radix*.

### 2.4. PPI Networks of Overlapping Targets Construction

The STRING database (https://cn.string-db.org/, accessed on 25 November 2022) was used to analyze protein–protein interactions (PPI) [28]. The intersection parts of *Astragali radix* active ingredient targets and periodontitis disease targets were entered into the STRING database to obtain the TSV file for constructing the PPI network. Further topological analysis was then performed using Cytoscape 3.9.0 software to adjust the size, shape, color and layout of the nodes to construct the complete PPI network and to elucidate the key regulatory proteins, and the top 10 nodes were filtered using the MCC algorithm with CytoHubba plugin.

### 2.5. GO Terms and KEGG Pathway Enrichment Analysis

Gene Ontology (GO) and Kyoto Encyclopedia of Genes and Genomes (KEGG) pathway enrichment analyses were performed using the Metascape platform (https://metascape.org/gp/index.html#/main/step1, accessed on 25 November 2022) to reflect the role of *Astragali radix* in the biological processes, molecular functions, cellular components and signaling pathways in the treatment of periodontitis [29]. Results were visualized on a bioinformatics platform (http://www.bioinformatics.com.cn/, accessed on 25 November 2022) [30]. Statistical significance was observed at *p* < 0.05, with a smaller *p*-value meaning higher enrichment and the larger count representing more genes being enriched [26,31]. Finally, the “*Astragali radix*-active ingredients-intersection targets and *Astragali radix*-active components-periodontitis-signaling pathway” network were constructed in Cytoscape 3.9.0 software.

### 2.6. Molecular Docking Simulation

Molecular docking technology can rapidly find drug targets and evaluate their pharmacological activities and molecular mechanisms, providing effective technical support for the modernization research of Chinese medicine [32]. It is of great significance for further understanding the interactions between compounds and their targets, as well as novel drug discoveries and understanding of molecular mechanisms. Molecular docking simulations are performed using Discovery Studio2019 software [33]. Discovery Studio2019 software (DS for short in below), normally applied in protein structure–function studies and drug discovery, provides an easy-to-use tool for protein simulation, optimization, and drug design. The ligand molecules in the complex were separated and then docked into the protein pocket of the complex, and the value of root-mean-square deviation (RMSD) was used to check the computational accuracy of the docking method. Normally, RMSD ≤ 2.0 Å is considered for the docking method to be accurate and the results to be reliable [34]. The CDOCKER module in DS software (v19.1.0.18287) is a CHARMM-based semi-flexible docking program for small molecule–protein docking, which mainly searches the flexible conformational space of ligand molecules by high-temperature kinetics and optimizes each conformation of the active site of the receptor by simulated annealing to make the docking results more accurate.

The top 10 key targets from the PPI network derived using the MCC algorithm with CytoHubba plugin, and the five core compounds obtained from the “*Astragali radix* active ingredients-intersection targets-periodontitis” network were molecularly docked as receptors and ligands. Two-dimensional (2D) structures of the active ingredient small molecules were obtained from the PubChem database (https://pubchem.ncbi.nlm.nih.gov/, accessed on 27 November 2022) and then mechanistic structure optimization was performed using Chem3D software (v20.0.0.41) [35]. Protein macromolecule structures were obtained from the PDB database (https://www.rcsb.org/, accessed on 27 November 2022) for their three-dimensional (3D) structures [36]. Then, the ligand and receptor were input into DS software (v19.1.0.18287).for molecular docking, and the docking score value called -CIE (-CDOCKER INTERACTION ENERGY) was calculated also. A larger -CIE value always means the better affinity of a small ligand with the 3D structure of a protein [37]. A higher molecular docking score suggests that the complex formed by the interaction of the active small molecule from *Astragali radix* with the target macromolecule associated with periodontitis exhibits greater stability, thereby enhancing its potential therapeutic effects.

### 2.7. Molecular Dynamics Analysis

Molecular dynamics (MD) simulations provide critical insights into the dynamic stability of receptor–ligand complexes under physiological conditions [26]. To verify the plausibility and stability of the docking results between a small molecule and the 3D structure of a protein, MD simulations were performed using GROMACS-2022 software (v5.0.4).package for the protein–ligand complexes that have the highest docking scores. Protein processing was carried out by using the CHARMM36 force field and small molecules were processed by using the Gaff2 force field. Using the TIP3 water model system, Na+ was added to balance the system charge. The solvent system was first energy minimized and thereafter the entire system was optimized with 5000 steps. The whole system was then preheated in NVT synthesis by adding harmonic constraints with a force constant size of 2 kcal/mol/Å2 to the protein backbone to ramp the system from 0 to 310 K in 100 ps. After the ramp was completed, the equilibration was first performed in NVT synthesis for 100 ps, setting the time step to 2 fs, to control the temperature and pressure, with the temperature being kept at 310 K and the pressure at 1 atm, respectively. Finally, the MM/GBSA method was used to calculate the binding free energy of the protein–ligand complex [38].

### 2.8. In Vitro Experimental Validation

#### 2.8.1. Cell Culture and Identification

Human periodontal ligament stem cells (hPDLSCs) were harvested from extracted wisdom teeth and identified using flow cytometry. This study was approved by the Biomedical Ethics Committee of Henan University (Kaifeng, China) on 26 March 2020 (Approval No: HUSOM2020-059). The hPDLSCs were cultured in Dulbecco’s Modified Eagle Medium/Nutrient Mixture F-12 (DMEM/F-12) supplemented with 10% fetal bovine serum (FBS) and 1% penicillin/streptomycin at 37 °C in an atmosphere containing 5% CO_2_. The cells were passaged during the logarithmic growth phase, achieving approximately 80% confluence. The cell density of hPDLSCs at passage 4 (P4) was adjusted to 1 × 10^6^ cells/mL, and the cells were divided into five 2 mL Eppendorf tubes, with 100 μL in each tube. The blank control group received no antibodies, while the four single-staining groups were treated with 5 μL of PE-labeled CD34, APC-labeled CD45, FITC-labeled CD90, and PE-Cyanine7-labeled CD105 antibodies (the four types of stem cell identification antibodies mentioned above are all sourced from Thermo Fisher Scientific Inc. (Waltham, MA, USA), 1:50 dilution in FACS buffer), respectively [36]. After thorough mixing, the samples were incubated at 4 °C for 30 min in the dark, followed by two washes with PBS, and finally, 200 μL of PBS suspension was added for flow cytometry machine (FongCyte™ S 4 Laser Flow Cytometer, Challen Biotechnology Co., Ltd., Beijing, China) detection.

#### 2.8.2. Cell Viability Assay

hPDLSCs were cultured in 96-well plates at a density of 5 × 10^3^ cells per well. After 24 h of cell attachment, the medium was replaced with various concentrations of Kaempferol (Ka, a known active component of *Astragali radix*) (0.01 μM, 0.1 μM, 1 μM, 10 μM, and 100 μM). The control group consisted of 0.1% DMSO, while the blank group contained medium without cells. Following 24 h of incubation, the old medium was discarded, and the cells were washed twice with fresh medium to eliminate drug interference. Subsequently, new medium containing 10% CCK-8 (Beyotime Biotechnology Co., Ltd. (Shanghai, China)) was added and incubated for 2 h. Finally, the absorbance at 450 nm was measured using a microplate reader.

#### 2.8.3. Live and Dead Cell Staining

hPDLSCs at a density of 5 × 10^5^ cells per well were inoculated into 24-well plates. Following a 48 h culture period, cells were then incubated with the working solution of the calcein-AM/PI double staining kit (Elabscience Biotechnology Co., Ltd. (Wuhan, China)) in a 37 °C incubator for 20 min. Subsequently, the cells were observed using a fluorescence microscope (Nikon, Tokyo, Japan) [39].

#### 2.8.4. Western Blot Analysis

Western blotting (WB) was employed to assess the expression levels of key target proteins, including IL-6, CASP3 and MMP9, in hPDLSCs [40]. The hPDLSCs were seeded into 6-well plates at a density of 1 × 10^6^ cells/mL and divided into four groups: control, LPS, Ka, and LPS + Ka, with three replicate wells for each group and LPS being at 85 μg/mL, concentration and Ka at 10μM, concentration. Briefly, after a 24 h culture period, total cellular protein was isolated with RIPA lysate. Then, an equal amount of total protein was separated using 10–12% sodium dodecyl sulfate polyacrylamide gel (SDS-PAGE) and subsequently transferred to a PVDF membrane. Following the membrane transfer, it was blocked with 5% skim milk at room temperature for 1 h, then incubated overnight with primary antibodies (IL-6, CASP3, MMP9, and GAPDH) at 4 °C (the four antibodies mentioned above were all purchased from Affinity Biosciences, Co., Ltd. (Liyang, China). The dilution ratios for the first three antibodies were 1:1000, while the dilution ratio for GAPDH was 1:5000). The membrane was washed three times with TBS-Tween 20 buffer and subsequently incubated with appropriate HRP-conjugated secondary antibodies at room temperature for 1 h. Finally, imaging was conducted using an exposure machine, with the addition of ECL luminescent reagent (Affinity Biosciences, Co., Ltd. (Liyang, China)) for exposure (the exposure device: MiniChemi 61 was purchased from Beijing Saizhi Entrepreneurial Technology Co., Ltd., Beijing, China) [41]. The grey values of each protein band were analyzed using ImageJ software (v1.54g), with GAPDH serving as the internal reference.

### 2.9. Statistical Analyses

Statistical analyses were performed using GraphPad Prism 9. Data were expressed as the mean ± standard deviation. Each cell experiment was conducted in triplicate. Statistical comparisons were carried out using the t-test or one-way analysis of variance (ANOVA).

## 3. Results

### 3.1. Screening and Identification of the Active Ingredients in Astragali radix and Their Corresponding Targets

From the TCMSP database, 87 components in *Astragali radix* were downloaded, and 17 active ingredients which have potential preventive and curative effects on periodontitis were obtained and confirmed based on the screening conditions of oral bioavailability (OB) ≥30% and drug-like property (DL) ≥0.18, and with proper target proteins in *Homo sapiens*, as shown in Table 1.

A total of 184 potential targets (without duplication) were collected, which were targeted by 17 herbal active ingredients, as shown in (Figure 1). It was hypothesized that there are complex pharmacological effects of *Astragali radix* in the treatment of a disease of *Homo sapiens*, these may relate to the 17 active ingredients mentioned above and their multiple corresponding targets [33].

### 3.2. Screening for Periodontitis-Related Targets

From the analysis of Gene Expression Omnibus (GEO, http://www.ncbi.nlm.nih.gov/geo/) based on two datasets, 280 differentially expressed genes related to periodontitis were isolated. The GSE10334 dataset included 89 up-regulated genes and 35 down-regulated genes, and the GSE16134 dataset included 123 up-regulated genes and 33 down-regulated genes. As shown in (Figure 2), heat and volcano maps were, respectively, plotted based on these screened differential genes, with up-regulated genes being indicated with red, and down-regulated genes being indicated with green [23]. To avoid missing some differential genes, we obtained 1880 differential genes from GeneCards, OMIM and DrugBank with the keywords of “periodontitis”, and 1996 genes related to periodontitis were confirmed based on the combination of the above three databases with GEO analysis and deletion of the duplicate genes (Figure 3A). The targets corresponding to the active ingredients in *Astragali radix* and the targets related to periodontitis were imported into the Venn diagram separately, and 94 intersection targets were obtained in total (Figure 3B).

### 3.3. Construction and Analysis of “Astragali radix-Active Ingredients-Targets-Periodontitis” Network

*Astragali radix*-active ingredient-target-periodontitis network was mapped and analyzed using Cytoscape 3.9.0 software. As shown in (Figure 4), most of the active ingredients in *Astragali radix* can be found to have close reactions with periodontitis-related targets.

### 3.4. Construction and Analysis of the PPI Network

The PPI network of intersecting targets was constructed and analyzed using Cytoscape 3.9.0 software to elucidate the potential mechanism of *Astragali radix* in the treatment of periodontitis. We calculated the degrees of all the nodes and arranged them in a circular layout based on their degree values, with larger node areas corresponding to higher degree values [15]. The network comprised 92 nodes and 3304 edges after excluding two isolated nodes. In (Figure 5A), target nodes with degree values exceeding 100 are depicted in pink and positioned in the inner layer. As illustrated in (Figure 5B), the top ten key targets were identified using the Degree algorithm with the CytoHubba plug-in [42], and these include TNF, IL6, AKT1, IL1β, VEGFA, TP53, PTGS2, CASP3, and EGF. These ten hub genes are recognized as core targets of the therapeutic effects of *Astragali radix* in the treatment of periodontitis.

### 3.5. Gene Function and Pathway Enrichment Analyses

GO enrichment analyses of the above 94 overlapping targets were performed by the Metascape platform with a setting of the threshold at *p*. adjust < 0.01. We found 19 biological process (BP)-related entries including responses to inorganic substances, positive regulation of cell migration, and responses to lipopolysaccharides; 20 molecular function (MF)-related entries including DNA-bound transcription factor binding, cytokine receptor binding, and kinase binding; and 19 cellular component (CC)-related entries including membrane rafts, transcriptional regulator complexes, and extracellular matrix. BP, MF and CC function analyses were shown with a bar graph in (Figure 6A). In addition, 146 signaling pathways were obtained from KEGG analyses with a threshold set at *p*. adjust < 0.01, and the top 20 major signaling pathways are presented in a bubble diagram [22], which mainly include pathways involved in cancer, lipids and atherosclerosis, AGE–RAGE signaling pathways in diabetic complications, and proteoglycans in cancer (Figure 6B).

The network of *Astragali radix*-active components-periodontitis targets-signaling pathways was constructed using the Cytoscape 3.9.0 software to elucidate the mechanism of *Astragali radix* in the treatment of periodontitis (Figure 7). From this network, it can be found that all 17 active ingredients in *Astragali radix* can target and modulate multiple pathways to produce therapeutic effects on periodontitis, especially the modulation of lipids and atherosclerosis, the AGE–RAGE signaling pathway in diabetic complications, and some cancer pathways.

### 3.6. Molecular Docking Analyses

A total of seven key targets were included in the molecular docking simulations (with a threshold at RSMD value ≤ 2.0 Å), namely AKT1, TNF, IL6, MMP9, PTGS2, CAPS3, and IL1β (Table 2), which are proteins known to be involved in inducing inflammation and/or cell death. Three other core targets, namely VEGFA, CXCL8 and IL10, were not analyzed with molecular docking because there are no suitable protein crystal structures in the PDB database [22].

The top five active components, such as Quercetin, Kaempferol, Formononetin, Calycosin and 7-O-methylisonicotinol were selected for molecular docking because they have more nodes and more edges in the network of “*Astragali radix*-active components-targets-periodontitis” (also shown in Figure 4). The screened active ingredients were prepared using the “Prepare Ligands” module to obtain a validated 3D conformation. After the removal of the crystalline water molecules, the “Prepare Protein” module was used to remove the multiple conformations of the target protein and to complement the incomplete amino acid residues [43]. Subsequently, molecular docking was performed in the “CDOCKER” module, which requires the use of -CIE to assess the affinity between protein and active ingredients, with the higher value meaning a better docking activity (Table 3).

The results for molecular docking of *Astragali radix* active ingredients with key targets are presented in (Figure 8). Quercetin (MOL000098) and 7-O-methylisomucronulatol (MOL000378) showed a good binding affinity with AKT1 and MMP9, while Kaempferol (MOL000422), Formononetin (MOL000392) and Calycosin (MOL000417) showed a good binding affinity with CASP3 and MMP9. Although these active ingredients have good binding affinity with AKT1, CAPS3 and MMP9, they also have high docking score values of -CIE with IL6, TNF, PTGS2, and IL1β, suggesting that these five active ingredients can bind stably with seven hub targets. In addition, we found that these active compounds interacted with the amino acid residues of core targets by forming hydrogen bonds, Pi–Pi bonds and van der Waals forces [44].

### 3.7. Molecular Dynamics Simulation

Furthermore, due to its highest docking score, the Kaempferol–CASP3 complex was analyzed further with molecular dynamics simulations to check the stability of their combination. RMSD relative to the initial position is often considered an important indicator of the stability of a ligand–target protein complex system, with the smaller value of the average RMSD always meaning the ligand combines more tightly with the target protein and the complex is more stable. The complex system first arrived at initial stability at around 2 ns, and then after some up and down fluctuations, it reached the final equilibrium state (<0.1 nm) at 24.5 ns after a large shock at 23 ns. These indicate that the binding of the Kaempferol–CASP3 gradually reaches final stability after 24.5 ns (Figure 9A).

The root mean square fluctuation (RMSF) can be used to track the fluctuation of the amino acid residues of a protein during the simulation, and it indicates the flexibility of local structure, with a smaller value of RMSF meaning that the complex system is more stable. While the overall structure of Kaempferol–CASP3 is relatively stable, the residues at positions 294 to 321 have large elasticity. Thus, the stability of the Kaempferol–CASP3 complex system may be enhanced after modifications at the above 27 residues (Figure 9B).

The radius of rotation (Rg) is an important indicator to assess the compactness of the structure. Although there were some fluctuations at the beginning, the Rg of the Kaempferol–CASP3 complex reached a peak at 24.5 ns and then kept at a relatively stable status (Figure 9C).

The solvent-accessible surface area (SASA) has a relationship with the solvation-free energy of molecules, which can be approximately proportional to their SASA with a completely hydrophobic surface. As shown in (Figure 9D), SASA was maintained at a relatively stable state within 50 ns, and this result was consistent with RMSD and RMSF values, indicating that there is a stable binding between Kaempferol and CASP3.

The stability of the system can be indirectly inferred from the number of hydrogen bonds. For example, after 24 ns, the number of hydrogen bonds increased significantly, indicating that the system was in a more stable state (Figure 9E). Finally, the three-dimensional structure of the Kaempferol–CASP3 complex is demonstrated in (Figure 9F).

The energy of the Kaempferol–CASP3 complex binding was calculated by using MM-PBSA [45], and the results are shown in (Figure 10A). GGAS normally represents the total gas-phase free energy and is calculated from the combination of (van der Waals energy) and EEL (electrostatic energy). The negative values (VDWAALS = −14.76, EEL = −9.22) of the Kaempferol–CASP3 complex binding indicate that both hydrophobic interactions and electrostatic interactions may have contributed to the stable binding. GSOLV includes the result of EGB (polar solvation storage energy,19.47 means it is harmful to the complex binding) and ESURF (nonpolar solvation storage energy, −2.38 means this energy is helpful for the binding of ligand and receptor protein).

The TOTAL energy (−6.89) was calculated by GGAS (−23.97) plus GSOLV (17.09), this negative TOTAL energy demonstrates that the Kaempferol–CASP3 complex binding is stable. As shown in (Figure 10B), the energy values of six residues (ARG:179 = −0.10, HIS:237 = −0.13, TYR:338 = −0.51, SER:339 = −0.16, TRP:340 = −1.41, and ARG:341 = −1.37) are all negative, indicating that Kaempferol mainly binding these five residues to develop a stable system.

### 3.8. Evaluation of the In Vitro Anti-Inflammatory Effects of Kaempferol

#### 3.8.1. Cultivation and Characterization of Stemness in hPDLSCs

Flow cytometry was employed to assess the expression of cell surface markers. As illustrated in (Figure 11A–D), the cells exhibited negative expression of hematopoietic-derived surface markers CD34 (0.72%) and CD45 (0.69%), while demonstrating positive expression of mesenchymal-derived surface markers CD90 (99.79%) and CD105 (99.60%). This indicates that the cultured hPDLSCs are mesenchymal-derived stem cells, suitable for subsequent experiments.

#### 3.8.2. Effect of Kaempferol on the Cell Viability

The effect of Kaempferol on the viability of hPDLSCs is illustrated in (Figure 11E). Compared to the control group, a Kaempferol concentration of 10 μM significantly enhanced cell viability (*p* < 0.05). The viability of hPDLSCs decreased by about 10% at a concentration of 100 μM compared to 10 μM; however, this difference was not statistically significant (*p* > 0.05). Thus, low concentrations of Kaempferol can promote the viability of hPDLSCs in a concentration-dependent manner. Since Kaempferol at the concentration of 10 μM was effective in promoting the proliferation of hPDLSCs; this concentration was chosen for subsequent experiments.

The results of fluorescence staining presented in (Figure 11F–H) indicate that after 24 h of cell culture, hPDLCs in the Kaempferol-treated group exhibited robust growth, characterized by fusiform cell shapes, prominent cell protrusions, and close intercellular connections, with no apparent dead cells observed. Under LPS induction, red fluorescence in the visual field was significantly enhanced, resulting in a substantial increase in dead cells (*p* < 0.05). However, following Kaempferol intervention, the intensity of red fluorescence was reduced, and the number of dead cells significantly decreased (*p* < 0.05). As shown in (Figure 11J), quantitative analyses further demonstrated that Kaempferol intervention significantly reduced LPS-induced inflammatory apoptosis in hPDLCs (*p* < 0.05).

#### 3.8.3. Kaempferol Significantly Inhibited the Protein Expression of IL6, CASP3 and MMP9 in LPS-Treated hPDLSCs

Based on our above studies in network pharmacology and molecular docking, we screened and predicted that Kaempferol treatment exhibits a high binding affinity for IL6, CASP3, and MMP9, suggesting that these proteins may be the primary targets of Kaempferol in the treatment of periodontitis. To further validate our findings, we assessed the expression of IL6, CASP3, and MMP9 proteins by WB analyses of treated cells. As illustrated in (Figure 12), the expression levels of IL6, CASP3, and MMP9 in both the control and Kaempferol alone-treated hPDLSCs were lower, whereas the levels in LPS-induced hPDLSCs were significantly higher than those in the groups without LPS induction. Kaempferol treatment significantly attenuated LPS-induced increased expression of IL-6, CASP3, and MMP9 (*p* < 0.05). These results are consistent with the findings from molecular docking and molecular dynamics simulations, indicating that Kaempferol ameliorates periodontitis at least through regulating the expression levels of three key proteins involved in inducing inflammation and cell death: IL6, CASP3, and MMP9.

## 4. Discussion

The etiology and pathogenesis of periodontitis are extremely complex. While plaque is known as an initiating factor, the bacterial–host interaction accelerates the development and progression of periodontitis [46]. However, although administration with antibiotics can reduce the pathogenic bacteria that invade periodontal tissues, the drugs are difficult to reach the periodontal pockets, and long-term usage of antibiotics can lead to drug resistance and some other systemic side effects [47]. TCM system mainly focuses on target therapy based on different clinical symptoms by using different herbal medicines. Since medicinal herbs have minimum toxic effects and many advantages, especially their abilities that control plaque, inhibit the growth of microorganisms, and promote tissue regeneration and anti-inflammation, some medicinal herbs have been proven to be suitable for periodontitis treatment [2]. In recent years, *Astragali radix* has been confirmed to alleviate alveolar bone destruction by regulating local inflammation and bone lysis during periodontitis [48]. However, the detailed molecular mechanisms remain unclear due to the presence of numerous components in *Astragali radix*. Modern bioinformatics analyses, including network pharmacology, molecular docking, molecular dynamics simulations, and in vitro experiments, can partially elucidate these mechanisms [49]. In the present study, we employed the aforementioned methods to investigate the mechanisms by which *Astragali radix* contributes to the treatment of periodontitis.

In this study, we found that Quercetin (MOL000098), Kaempferol (MOL000422), Formononetin (MOL000392), Calycosin (MOL000417), and 7-O-methylisomucronulatol (MOL000378) are the most important components in *Astragali radix* in treating periodontitis, and Quercetin, with the highest degree value, is the most critical active component in *Astragali radix*. Quercetin, a natural flavonoid, has already been proven to have anti-inflammatory, antibacterial and antioxidant effects, which can kill *P. gingivalis* and inhibit a variety of virulence factors [50]. Kaempferol, also an important component of *Astragali radix*, inhibited the release of TNF-α, increased the expression and secretion of IL-10, and exerted significant anti-inflammatory effects [51,52]. In experimental models of periodontitis in rats, Kaempferol reduced the alveolar bone resorption, inhibiting the attachment loss and MMP-1 production [53]. In periodontal ligament stem cells (PDLSCs) cultured in vitro, it also enhanced their proliferation and osteogenesis (bone formation potential) by activating the Wnt/beta-catenin signaling pathway, suggesting that Kaempferol could have a potential clinical application for periodontal tissue regeneration [54]. In addition, Formononetin, an active ingredient in *Astragali radix*, also showed therapeutic effects in preventing periodontal disease [55].

By using the Degree algorithm of the CytoHubba plugin, among the 92 targets of the PPI network constructed, the top 10 targets which are also considered core targets were obtained, including TNF, IL1β, IL6, PTGS2 (also known as COX-2), TP53, MMP9, EGF, AKT1, VEGFA, and CASP3. These core target proteins are involved in immune regulation, inflammatory response, angiogenesis, oxidative stress, cell proliferation and apoptosis. TNF-α, mainly generated in inflammation reactions, can cause continuous production of osteoclasts (bone resorptive cells) and increase their resorption activity, thus leading to the loss of alveolar bone [56]. IL6 is one of the most potent proinflammatory cytokines and is involved in the periodontitis occurrence in human gingival fibroblasts cultured in vitro [57]. In addition, MMP-9 was shown to be a sensitive marker for periodontal inflammation during orthodontic treatment [58,59], and VEGF-A, which belongs to the vascular endothelial growth factor (VEGF) family, plays an important role during the pathological process of periodontitis [60]. Similarly, CASP3, which mainly regulates cell apoptosis, is also involved in the pathogenesis of periodontal disease [61]. Since all the above-mentioned genes are known to participate in the development of periodontitis as they have already been confirmed by other researchers, our results of identified key intersection targets are reliable.

To help us clarify which biological processes, cellular components, and molecular functions are involved in the therapy of periodontitis by *Astragali radix*, the current study has undergone GO analyses. In addition, we undertook KEGG analyses to help us to find out which pathways participate in this process. Our analyses suggest that, during the treatment of periodontitis, *Astragali radix* active ingredients mainly involve the following signaling pathways, including pathways in cancer, and the AGE–RAGE signaling pathway involved in diabetic complications and proteoglycan metabolism in cancer. Oxidative stress and lipid peroxidation are involved in various pathological states, including inflammation, atherosclerosis, neurodegenerative diseases, cancer and also periodontitis [62]. In a rat model of ligation-induced periodontitis, increased lipid peroxidation was found in serum and aorta as well as in periodontal tissue [63]. Periodontitis is a risky factor for the development of atherosclerosis because periodontitis-induced aortic lipid peroxidation which is closely associated with the early stages of atherosclerosis [63]. The AGE–RAGE signaling pathway is associated with the regulation of oxidative stress and inflammation [64], and RAGE was found strongly expressed in the gingiva of periodontitis patients with or without diabetes [64]. In gingival fibroblasts cultured in vitro, the accumulation of AGEs may upregulate the expression of MMP-1, and the RAGE/NF-κB pathway is involved in the metabolism of MMP-1, and thus both pathways may be important for the development of diabetes-associated periodontitis [65]. Based on our results of GO and KEGG analyses, we hypothesize that *Astragali radix* treats periodontitis at least by regulating the accumulation of late glycosylation end products, oxidative stress response, inflammatory response and atherosclerosis, and eventually reducing the inflammation in periodontitis.

In addition, our molecular docking analysis results are consistent with our network pharmacology analysis results. It is worth noting that the complex of Kaempferol–CASP3 achieved the highest binding score (-CIE = 72.18), and thus it should have a stable binding structure. These results were confirmed by molecular dynamics simulations which are consistent with the results of the molecular docking, and it was proven that interactions such as van der Waals forces and hydrogen bonds play important roles in the stability of the complex. Therefore, Kaempferol, which targets CASP3 to inhibit cell apoptosis, can also partly explain the pharmacological effect of *Astragali radix* in the treatment of periodontitis.

Furthermore, by conducting some in vitro experiments with cultured human periodontal ligament stem cells (hPDLSCs) (known as a promising source of cells for regenerative therapy in periodontitis), we validated that Kaempferol primarily ameliorates periodontitis by modulating inflammation and cell death. Our results demonstrated that low concentrations of Kaempferol exhibited no toxicity to hPDLSCs and significantly mitigated LPS-induced cellular damage. Furthermore, cellular experiments confirmed that Kaempferol, the most prevalent compound of *Astragali radix*, reduced the expression of IL6, CASP3, and MMP9, factors known to induce inflammation and cell death. indicating that Kaempferol exerts anti-inflammatory effects by binding to target proteins and modulating their expression.

Nonetheless, our study has certain limitations. First, we searched four commonly used databases to maximize the comprehensiveness of our collection of disease-related targets; however, we acknowledge that this approach may not encompass all results, potentially leaving out some targets and limiting the interpretation of our findings. Second, our pharmacological analyses indicated that it is possible that multiple active compounds of *Astragali radix* may exert a synergistic effect through various targets and signaling pathways. Third, this study employed bioinformatics to identify the key active components of *Astragali radix* relevant to the treatment of periodontitis, including compounds such as kaempferol and quercetin. However, due to the constraints of one private manuscript length, it was not feasible to comprehensively present all the in vitro experimental results of other significant active components of *Astragali radix* related to periodontitis, such as quercetin and formononetin, which also have notable bioactivities. Finally, the in vitro experimental section of this study primarily utilized conventional experiments such as stem cell identification, cell viability assessment, and protein immunoblotting. However, there remains a lack of studies on in vitro gene knockout models and protein–protein interactions analysis. In subsequent experiments, it is necessary to further refine the relevant research, with the aim of elucidating the anti-periodontitis effects of kaempferol from a more in-depth perspective. Further studies are needed to address these limitations.

## 5. Conclusions

Based on preliminary validations obtained from bioinformatics analyses and in vitro experiments, we have drawn the following conclusions: First, *Astragali radix* exerts its anti-periodontitis effects through various components, including kaempferol and quercetin, which act on multiple targets associated with periodontitis, such as IL-6 and MMP-9. Second, additional validation through in vitro experiments indicated that kaempferol significantly reduces the expression levels of inflammatory factors, including IL-6, CASP3, and MMP-9, in LPS-induced human periodontal ligament stem cells.

The above findings provide a promising basis for the targeted use of *Astragali radix* in periodontitis. However, given that kaempferol is a flavonoid with low oral bioavailability and presents challenges in achieving effective concentrations following systemic administration, there is a need for suitable carriers to facilitate the development of localized treatments. Future studies could explore the potential of combining kaempferol with hydrogels, electrospinning, or 3D-printed scaffolds to develop localized or systemic applications. Nonetheless, we continue to encounter numerous challenges in conducting more comprehensive preclinical experiments, which still need a long way to go.

## Figures and Tables

**Figure 1 nutrients-17-00627-f001:**
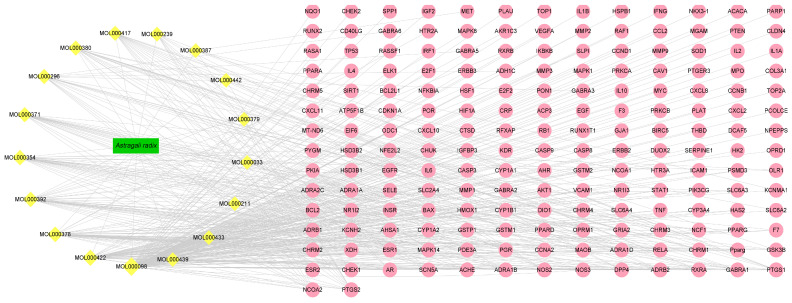
*Astragali radix* active ingredients and the related target network. The yellow diamond-shaped blocks represent the active components of *Astragali radix*. The green rectangular blocks represent the *Astragali radix* plant itself. The pink circles denote the targets associated with the various active components of *Astragali radix* which relate to human physiological functions.

**Figure 2 nutrients-17-00627-f002:**
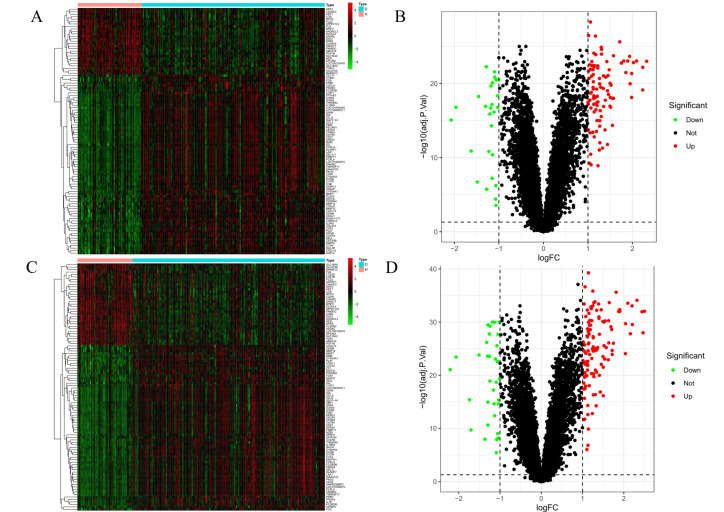
Heat and volcano maps of differentially expressed genes. (**A**) Heat map based on GSE10334; (**B**) Volcano map based on GSE10334; (**C**) Heat map based on GSE16134; (**D**) Volcano map based on GSE16134. In the volcano plot and heatmap, red denotes the upregulated genes associated with the onset and progression of periodontitis. Conversely, green signifies the downregulated genes associated with the progression of periodontitis.

**Figure 3 nutrients-17-00627-f003:**
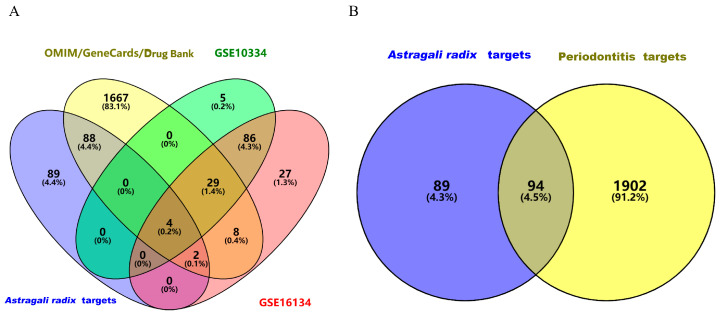
Venny map. (**A**) The integration of the active ingredient targets of *Astragali radix* and the targets related to periodontitis across four databases; (**B**) *Astragali radix* active ingredient targets and periodontitis-related targets and their overlapping.

**Figure 4 nutrients-17-00627-f004:**
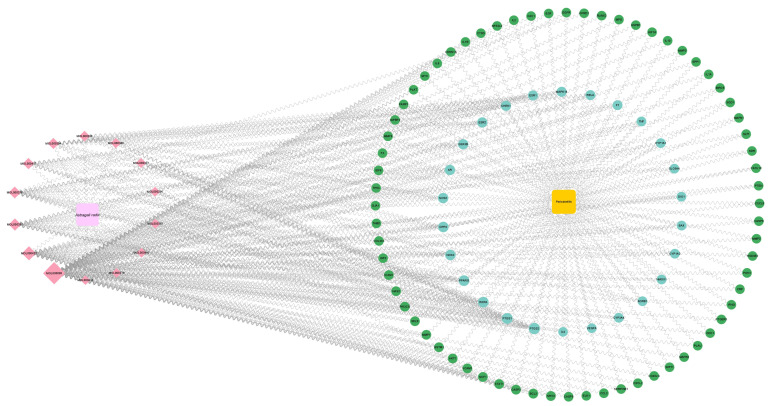
Construction and analysis of the network of “Astragali radix-active ingredi-ents-targets-periodontitis”. The light pink square symbolizes Astragali radix. The dark pink dia-mond denotes the active components of Astragali radix that engage with targets associated with periodontitis. The yellow square symbolizes periodontitis. The green and blue circles denote the periodontitis-related targets that interact with the active components of Astragali radix.

**Figure 5 nutrients-17-00627-f005:**
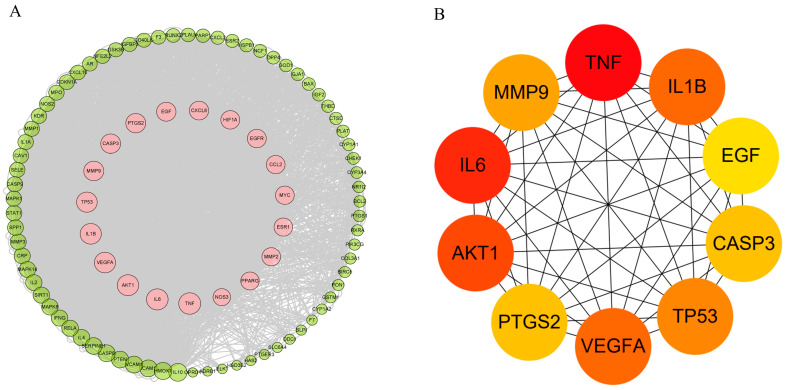
PPI network map. (**A**) A diagram illustrating the network of Astragali radix, active components, targets, and periodontitis (The larger the area of the circle, the higher the degree values); (**B**) The core target network (The color of the nodes transitions from yellow to red, indicating a higher ranking among key genes).

**Figure 6 nutrients-17-00627-f006:**
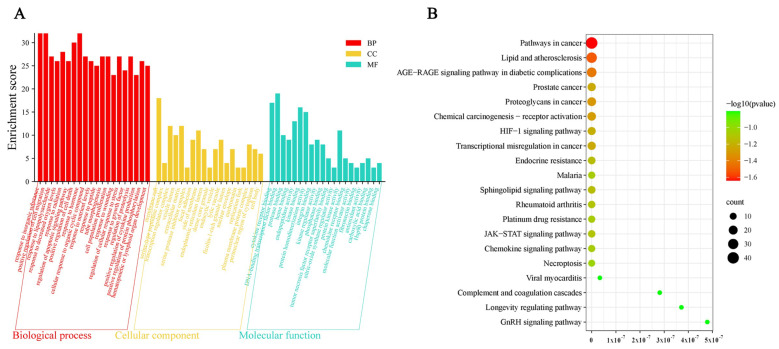
GO functional analysis and KEGG pathway enrichment analyses. (**A**) Outcomes of entries related to biological processes (BP), cellular components (CC) and molecular functions (MF) from the GO enrichment analysis (The red columns represent biological processes, the yellow columns indicate cellular components, and the blue columns denote molecular functions). (**B**) The top 20 pathways from the pathway enrichment analysis (The transition of color from green to red, accompanied by an increase in node area from small to large, indicates an increase in the level of enrichment).

**Figure 7 nutrients-17-00627-f007:**
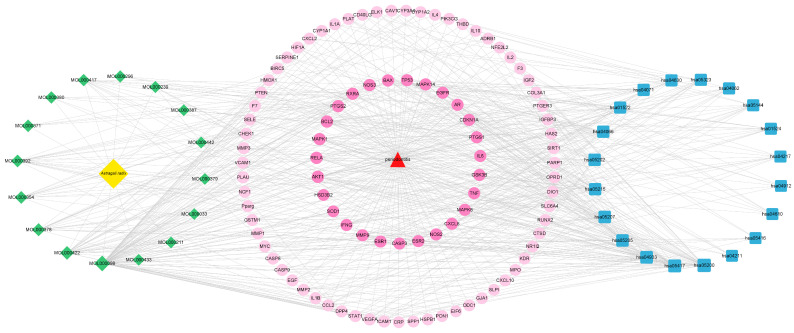
*Astragali radix* active component-periodontitis-target-signaling pathway network (The yellow diamond symbolizes *Astragali radix*, the green diamond signifies the active components of *Astragali radix* that engage with targets associated with periodontitis, the red triangle denotes periodontitis, the pink circle illustrates the periodontitis-associated targets that interact with the active components of *Astragali radix*, and the blue square depicts the signaling pathways enriched by the aforementioned periodontitis targets).

**Figure 8 nutrients-17-00627-f008:**
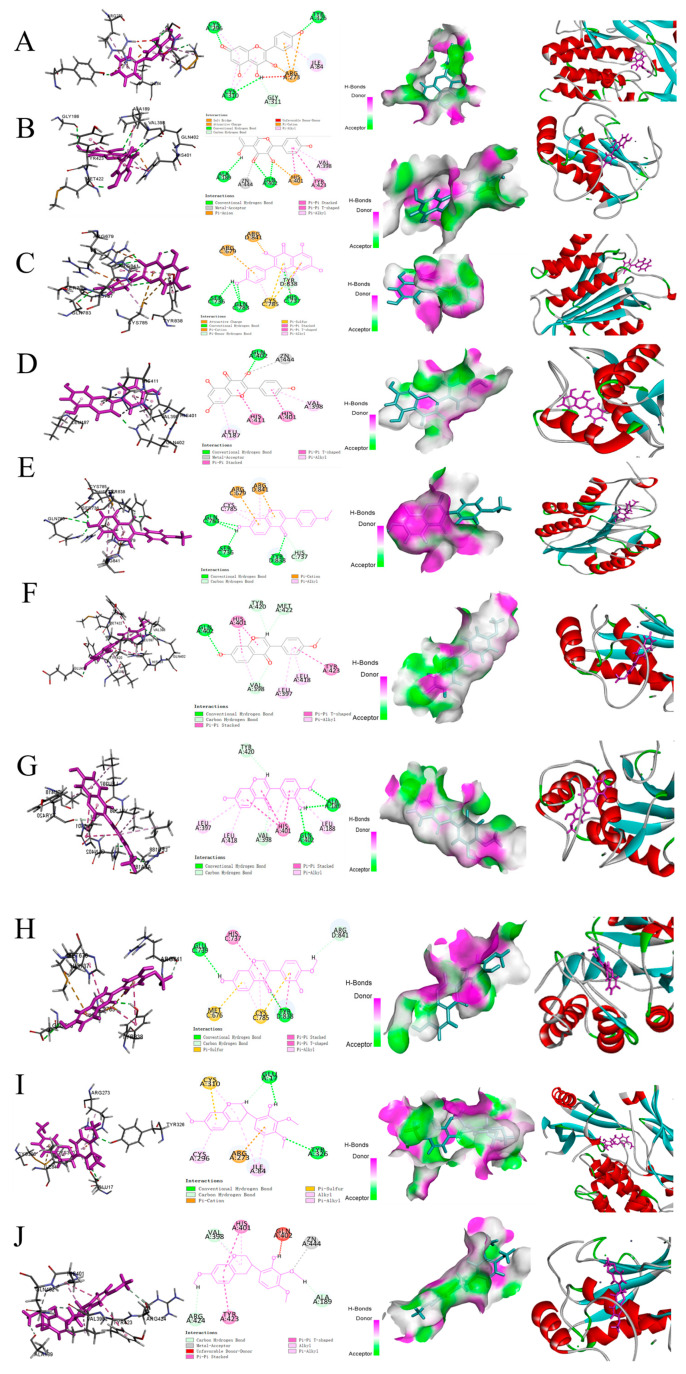
A display of results for the top 10 best molecular docking. (**A**) Quercetin-AKT1 (PDB ID: 4EJN); (**B**) Quercetin-MMP9 (PDB ID: 2OW1); (**C**) Kaempferol-CAPS3 (PDB ID: 1RHM); (**D**) Kaempferol-MMP9 (PDB ID: 2OW1); (**E**) Formononetin-CAPS3 (PDB ID: 1RHM); (**F**) Formononetin-MMP9 (PDB ID: 20W1); (**G**) Calycosin-MMP9 (2OW1); (**H**) Calycosin-CAPS3 (1RHM); (**I**) 7-O-methylisomucronulatol-AKT1 (4EJN); (**J**) 7-O-methylisomucronulatol-MMP9 (2OW1). The results presented above, arranged from left to right, comprise the amino acid residue diagram, the two-dimensional structural representation of the complex, the hydrogen bond interaction diagram, and the protein pocket illustration.

**Figure 9 nutrients-17-00627-f009:**
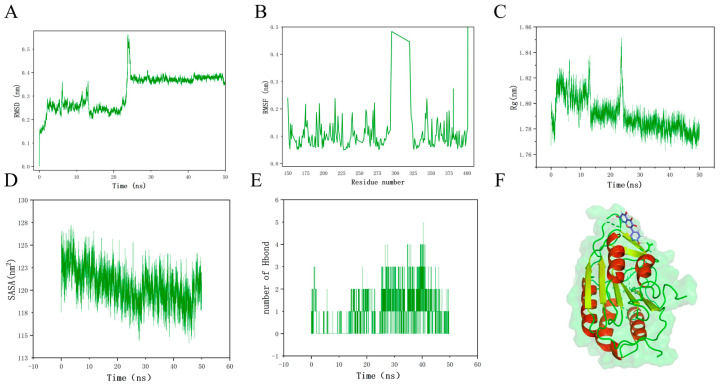
Results of molecular dynamics simulation of the CASP3—kaempferol complex. (**A**) RMSD; (**B**) RMSF; (**C**) Rg; (**D**) SASA; (**E**) Number of hydrogen bonds; (**F**) 3D structure of CASP3—kaempferol complex.

**Figure 10 nutrients-17-00627-f010:**
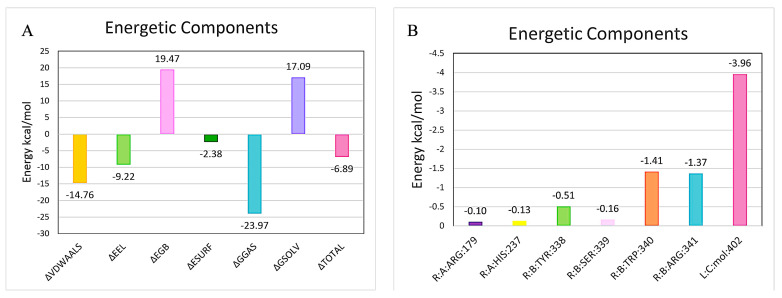
Molecular dynamics simulation energy analyses for the Kaempferol–CASP3 complex. (**A**) Binding energy of the protein–ligand complex. VDWAALS: van der Waals energy; EEl: Electrostatic energy; GGAS: Total gas phase free energy. EGB: Polar solvation energy; ESURF: Non-polar solvation energy; GSOLV: Total solvation free energy. TOTAL: GSOLV + GGAS; (**B**) The relationship of each contact residue to the binding.

**Figure 11 nutrients-17-00627-f011:**
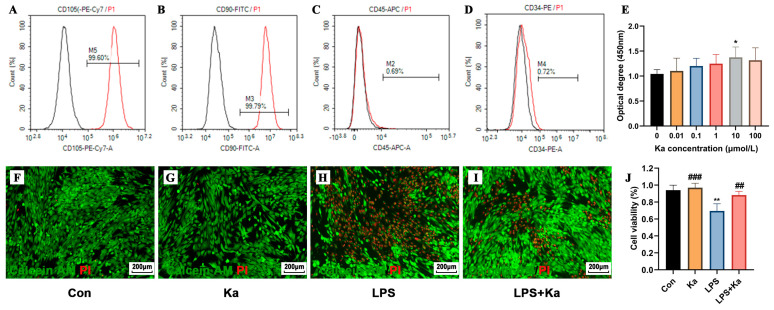
Stem cell marker characterization and the effects of Kaempferol (Ka) treatment on the viability of hPDLSCs. (**A**–**D**) The results of stem cell identification in hPDLSCs. (**E**) Dose-dependent effect of Ka on cell viability as measured by CCK8 assays. (**F**–**I**) Effects of Ka with/without insead of “±”. LPS treatment on cell viability and cell death as shown by fluorescent staining (viable cells, green staining; apoptotic cells, red staining). (**J**) Quantitative analyses of living and dead cells. ** *p* < 0.01 vs. Con, * *p* < 0.05 vs. Con, ### *p* < 0.001 vs. LPS group (LPS), ## *p* < 0.01 vs. LPS.

**Figure 12 nutrients-17-00627-f012:**
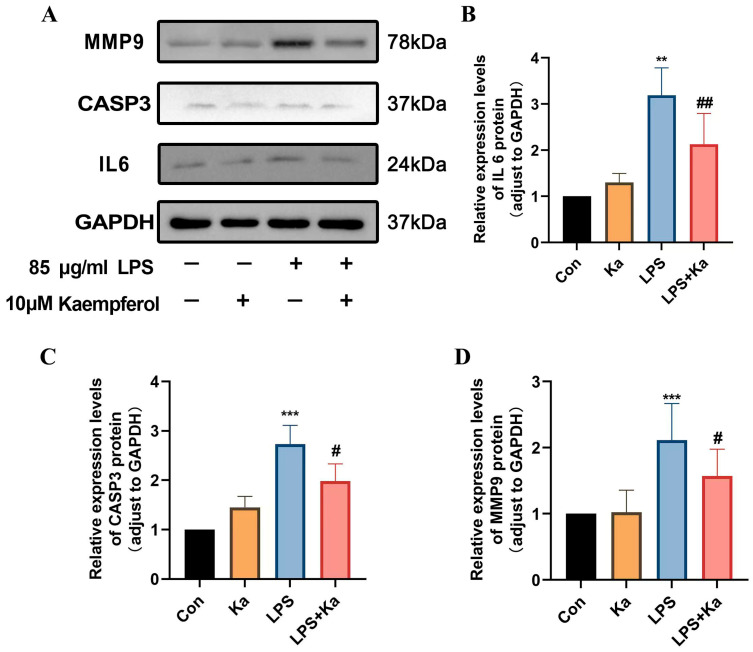
Effects of Kaempferol (Ka) treatment on viability and LPS-induced cell death in hPDLSCs. (**A**) Western blotting analyses of protein expression of IL6, CASP3, and MMP9. (**B**–**D**) Quantitative analysis of IL6, CASP3 and MMP9 proteins. *** *p* < 0.001 vs. control group (Con), ** *p* < 0.01 vs. Con, ## *p* < 0.01 vs. LPS, and # *p* < 0.05 vs. LPS.

**Table 1 nutrients-17-00627-t001:** Active compounds in *Astragali radix*.

Number	Molecular ID	Molecule Name	OB (%)	DL
HQ1	MOL000211	Mairin	55.38	0.78
HQ2	MOL000239	Jaranol	50.83	0.29
HQ3	MOL000296	hederagenin	36.91	0.75
HQ4	MOL000033	(3S,8S,9S,10R,13R,14S,17R)-10,13-dimethyl-17-[(2R,5S)-5-propan-2-yloctan-2-yl]-2,3,4,7,8,9,11,12,14,15,16,17-dodecahydro-1H-cyclopenta[a]phenanthren-3-ol	36.23	0.78
HQ5	MOL000354	Isorhamnetin	49.60	0.31
HQ6	MOL000371	3,9-di-O-methylnissolin	53.74	0.48
HQ7	MOL000378	7-O-methylisomucronulatol	74.69	0.30
HQ8	MOL000392	Formononetin	69.67	0.21
HQ9	MOL000422	Kaempferol	41.88	0.24
HQ10	MOL000433	FA	68.96	0.71
HQ11	MOL000439	isomucronulatol-7,2′-di-O-glucosiole	49.28	0.62
HQ12	MOL000442	1,7-Dihydroxy-3,9-dimethoxy pterocarpene	39.05	0.48
HQ13	MOL000098	Quercetin	46.43	0.28
HQ14	MOL000379	9,10-dimethoxypterocarpan-3-O-β-D-glucoside	36.74	0.92
HQ15	MOL000380	(6aR,11aR)-9,10-dimethoxy-6a,11a-dihydro-6H-benzofurano [3,2-c]chromen-3-ol	64.26	0.42
HQ16	MOL000387	Bifendate	31.10	0.67
HQ17	MOL000417	Calycosin	47.75	0.24

Oral bioavailability (OB); drug-like property (DL).

**Table 2 nutrients-17-00627-t002:** Molecular information of key target proteins.

Target	PDB ID	Ligand	Resolution/Å	RSMD/Å
IL6	1ALU	TLA	1.90	1.6702
TNF	7KP9	ATG	2.15	0.9369
IL1β	5R88	LWA	1.38	1.4982
AKT1	4EJN	OR4	2.19	0.4065
MMP9	2OW1	TMR	2.20	0.2574
PTGS2	5IKV	FLF	2.51	0.7210
CAPS3	1RHM	NA4	2.50	1.8978

Root-mean-square deviation (RMSD), Protein Data Bank (PDB).

**Table 3 nutrients-17-00627-t003:** Molecular docking of key target proteins and active compounds.

Name	Molecular Docking Score for Key Targets (-CIE)						
	IL1β	TNF	AKT1	IL 6	PTGS2	MMP9	CASP3
Quercetin	50.51	42.73	55.74	41.66	46.45	53.66	52.52
Kaempferol	47.88	58.51	60.21	41.97	59.20	61.39	72.18
Formononetin	41.04	47.03	45.73	24.78	44.23	50.55	50.75
Calycosin	43.58	48.78	47.92	27.56	51.22	54.67	52.29
7-O-methylisomucronulatol	33.62	43.10	44.95	26.70	27.64	47.23	28.28

-CIE (-CDOCKER interaction energy).

## Data Availability

Data will be made available upon request.

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
