# Peer review of "The Multi-Target Action Mechanism for the Anti-Periodontitis Effect of Astragali radix Based on Bioinformatics Analysis and In Vitro Verification"

_nutrients, 2025, doi:10.3390/nu17040627_

Round 1
Reviewer 1 Report
Comments and Suggestions for Authors
The manuscript accounts of an interesting study aimed to reveal the molecular mechanisms by which Astragali radix exerts its therapeutic effect on periodontitis, using employ network pharmacology, molecular docking, molecular dynamic analysis but also in vitro experiments on human periodontal ligament stem cells. The methodology is modern. Results of in silico studies are combined with experimental results, confirming a part of in silico predictions. The presentation is generally correct (remarks are presented below). The discussion of the limitations of the study is well written and includes the main reservations one could have with respect to the presented results.
Remarks
Line 59: What does it mean „warm nature”
Lines 194/195: Please provide some data on the donors of the wisdom teeth.
Line 195: How the stem cells were identifies? Please provide details.
Lines 211and next: “μM/L” is not a proper unit, since, by definition μM= μmol/L, so please use “μM” or “μmol/L”.
Line 230: Please explain the acronym.
Section 2.8.4. Please provide the source and dilution of antibodies, identify the ECL luminescent reagent and the exposure machine.
Line 261: “homo”, please correct to “Homo”
Lines 438/439: it could be added that the cell viability at 100 μM Ka (though lower than at 100 μM Ka) was not decreased with respect to the control.
Figure 12 is lacking in the manuscript. The legend does not correspond to what is designated as Fig. 12 in the Supplementary Material.
Line 601: “to reveal the main mechanisms”, I would be more careful and suggest “to reveal the main possible mechanisms”, taking into account the discussion of the limitations of the study. The same refers to Line 79.
References should be formatted according to the MDPI requirements.
Author Response
Point to Point Response
Many thanks for the kindhearted review and constructive comments from reviewer. We revised the manuscript carefully, and hope we already addressed all the concerns of editor this time. Please see our point-to-point response which shown in below.
Reviewer 1
Comment 1: The manuscript accounts of an interesting study aimed to reveal the molecular mechanisms by which Astragali radix exerts its therapeutic effect on periodontitis, using employ network pharmacology, molecular docking, molecular dynamic analysis but also in vitro experiments on human periodontal ligament stem cells. The methodology is modern. Results of in silico studies are combined with experimental results, confirming a part of in silico predictions. The presentation is generally correct (remarks are presented below). The discussion of the limitations of the study is well written and includes the main reservations one could have with respect to the presented results.
Answer 1: Thank you for your comments.
Comment 2: Line 59: What does it mean, warm nature”
Answer 2: Thank you for your comments. As indicated in the reference (Zhang, Y., Chin Med, 2024), according to the principles of traditional Chinese medicine, the properties of herbs can be classified into four categories: cold, hot, warm, and cool, collectively known as the “Four Natures.” Warm-natured herbs are recognized for their ability to strengthen the liver, enhance bone health, and boost immunity. Astragali radix is categorized as a warm-natured herb.
Comment 3: Lines 194/195: Please provide some data on the donors of the wisdom teeth.
Answer 3: Thank you for your question. Regarding the information you mentioned about the wisdom tooth donors, we will provide a copy of the informed consent form signed by the wisdom tooth donors for your reference, which includes information such as the donor’s name, age, gender, and whether they have any systemic diseases. For specific details, please refer to the informed consent form.
Comment 4: Line 195: How the stem cells were identifies? Please provide details.
Answer 4: Thank you for your suggestions. Based on the previous research published by our group (Feng, J., Heliyon 2024), the following conclusions can be drawn: the characteristics of human periodontal ligament stem cells derived from mesenchyme are positive for CD90 and CD105, while CD34 and CD45 are negative. The results of this study are consistent with the aforementioned literature. Relevant content has been added to the manuscript, with modifications labelled in blue.
Comment 5: Lines 211and next: “μM/L” is not a proper unit, since, by definition μM= μmol/L, so please use “μM” or “μmol/L”.
Answer 5: Thank you very much for your suggestions. We have made modifications to the relevant content in the whole manuscript, and the changes have been labelled in blue color.
Comment 6: Line 230: Please explain the acronym.”.
Answer 6: Thank you for your valuable suggestions. Regarding the term "acronym," we conducted a systematic and thorough search in the manuscript text but did not find its occurrence. Furthermore, we meticulously checked the spelling of all words in this paper to ensure accuracy.
Comment 7: Section 2.8.4. Please provide the source and dilution of antibodies, identify the ECL luminescent reagent and the exposure machine.
Answer 7: We appreciate your valuable suggestions. The manuscript now includes detailed information about the sources and dilution ratios of the antibodies utilized for protein immunoblotting in the in vitro experiments, as well as the ECL solution and exposure equipment. The modified content has been clearly labelled in blue.
Comment 8: Line 261: “homo”, please correct to “Homo”.
Answer 8: Thank you for your valuable suggestions. The relevant content has been revised in the manuscript. The modified sections have been labelled in blue.
Comment 9: Lines 438/439: it could be added that the cell viability at 100 μM Ka (though lower than at 100 μM Ka) was not decreased with respect to the control.
Answer 9: Thank you for your valuable suggestions. We have made appropriate revisions in the relevant sections of the paper according to your suggestions. The modified sections have been labelled in blue.
Comment 10: Figure 12 is lacking in the manuscript. The legend does not correspond to what is designated as Fig. 12 in the Supplementary Material.
Answer 10: Thank you for your comments. We carefully reviewed and ensured the accuracy of the figures in the manuscript at the time of the initial submission. Regarding your mention of the missing Figure 12 in the manuscript, we speculate that this may have occurred due to the large number of figures, which led to an oversight during the editing and formatting process. We have included Figure 12 in this revised version which has been resubmitted.
Comment 11: Line 601: “to reveal the main mechanisms”, I would be more careful and suggest “to reveal the main possible mechanisms”, taking into account the discussion of the limitations of the study. The same refers to Line 79.
Answer 11: Thank you for your suggestions This study still has some limitations, as it only employed conventional methods such as cell viability assays and Western blotting for validation, lacking experimental verification through gene knockout models and in vitro protein-protein interaction detections. Therefore, this study is expected to reveal the primary mechanisms by which Astragali radix exerts its effects against periodontitis. Relevant content has been appropriately revised in the main text, with the modifications labelled in blue.
Comment 12: References should be formatted according to the MDPI requirements.
Answer 12: We sincerely appreciate your valuable suggestions. We have revised the formatting of all references in the manuscript to align with the MDPI journal’s requirements, thereby ensuring compliance with the journal’s publication standards.

Reviewer 2 Report
Comments and Suggestions for Authors
The authors present an in silico pharmacology approach to identify potential active compounds against proteins associated with periodontitis from Astragali radix. Overall, the work is well presented, showcasing how computational approaches can guide in vitro experiments towards drug research (or, as possible in this case, towards wellness products). What is missing though are the results from the in vitro studies for all the identified compounds and possible mixtures as they are within Astragali radix. The authors should explain why only Kaempferol in vitro data are shown and not for Quercetin, Formononetin, Calycosin, or 7-O-methylisomucronulatol. Although Kaempferol could be the most potent compound, as network analysis reveals, the herb could provide its pharmacological activity through synergies of all identified compounds. Otherwise, why use the herb and not only one compound?
Some additional comments.
1) Figure resolution and clarity.
2) I would argue to include phrases such as "the precise development" (line 32).
Author Response
Point to Point Response
Many thanks for the kindhearted review and constructive comments from reviewer. We revised the manuscript carefully, and hope we already addressed all the concerns this time. Please see our point-to-point response which shown in below.
Reviewer 2
Comment 1: The authors present an in silico pharmacology approach to identify potential active compounds against proteins associated with periodontitis from Astragali radix. Overall, the work is well presented, showcasing how computational approaches can guide in vitro experiments towards drug research (or, as possible in this case, towards wellness products). What is missing though are the results from the in vitro studies for all the identified compounds and possible mixtures as they are within Astragali radix. The authors should explain why only Kaempferol in vitro data are shown and not for Quercetin, Formononetin, Calycosin, or 7-O-methylisomucronulatol. Although Kaempferol could be the most potent compound, as network analysis reveals, the herb could provide its pharmacological activity through synergies of all identified compounds. Otherwise, why use the herb and not only one compound?
Answer 1: We sincerely appreciate your constructive feedback on our study. We identified the primary active components of Astragali radix, including kaempferol and quercetin, which exhibit significant effects against periodontitis by using network pharmacology analysis. Preliminary validation was performed utilizing molecular docking and molecular dynamics simulation techniques. In subsequent in vitro experiments, we concentrated on validating the anti-periodontitis effects of kaempferol. Due to the constraints imposed by the length of one manuscript, we were unable to comprehensively present the in vitro experimental results of other significant active components of Astragalus against periodontitis, including quercetin and isorhamnetin, although they also represent notable bioactivities. Related research is currently still ongoing, and we will report on these findings in our future studies. The relevant limitations of our manuscript have been revised in the manuscript and are labelled in blue color.
Comment 2: Some additional comments :1) Figure resolution and clarity.
Answer 2: Thanks for your comments. All images in this study have been modified and examined to ensure that each image possesses high clarity and resolution.
Comment 3: Some additional comments: 2) I would argue to include phrases such as “the precise development” (line 32).
Answer 3: We sincerely appreciate your valuable suggestions. In response to the frequent use of phrases such as “the precise development” in the text, we have made comprehensive and meticulous revisions to ensure that these changes do not affect the original structure of the paper. All modifications have been labelled in blue.

Reviewer 3 Report
Comments and Suggestions for Authors
Dear Authors,
The manuscript titled " The multi-target action mechanism for the anti-periodontitis effect of Astragali radix based on bioinformatics analysis and in vitro verification" explores the therapeutic potential of Astragali radix in periodontitis treatment. The study highlights its anti-inflammatory and anti-apoptotic mechanisms, focusing on Kaempferol as a key compound and identifying seven core targets (e.g., IL6, TNF, CASP3). Molecular docking and in vitro experiments confirm its ability to inhibit LPS-induced apoptosis and reduce inflammatory marker expression. These findings provide a promising basis for the targeted use of Astragali radix in periodontitis.
Several critical aspects need attention to improve its clarity, coherence, and overall impact
Introduction
Strengths:
- Provides a solid background on periodontitis and the relevance of Astragali radix in Traditional Chinese Medicine (TCM).
- Justifies the study by highlighting gaps in current research on the molecular mechanisms of Astragali radix.
Critical Issues:
- Redundant explanations about TCM and periodontitis could be trimmed.
- The link between bioinformatics approaches and traditional medicine could be better articulated.
- The novelty of the research is implied but not explicitly stated.
Methods and Materials
Strengths:
- Thorough explanation of methodologies such as network pharmacology, molecular docking, and in vitro experiments.
- Comprehensive descriptions of software and databases used.
Critical Issues:
- Some sections, such as molecular docking and dynamics simulations, are overly technical and may not be accessible to a broader audience.
- Statistical methods need further clarification, particularly how key thresholds (e.g., p-values) were chosen.
- No mention of reproducibility or potential limitations of the computational models.
Results
Strengths:
- Systematic presentation of findings with appropriate use of tables, figures, and networks.
- The progression from identifying active ingredients to validating their molecular effects is logical.
Critical Issues:
- Certain figures and tables (e.g., heatmaps, docking results) are overly complex and lack concise legends for easier interpretation.
- Overemphasis on numerical values (e.g., docking scores, p-values) without sufficient biological interpretation.
- The narrative often shifts between technical details and broader implications without clear transitions.
Discussion
Strengths:
- Adequate integration of results with previous literature to support conclusions.
- Provides insights into the therapeutic potential of Kaempferol and other active compounds.
Critical Issues:
- The discussion is lengthy and sometimes reiterates results rather than interpreting them.
- Potential clinical applications are mentioned but not deeply explored.
- Limitations of the study are acknowledged but could be expanded to include the generalizability of in vitro findings.
Conclusion
Strengths:
- Summarizes the main findings effectively.
- Reiterates the potential of Astragali radix as a therapeutic agent for periodontitis.
Critical Issues:
- The conclusion is overly general and lacks specific actionable recommendations for future research.
- The statement on “scientific evidence for related drug development” could benefit from clarification or examples.
References
Strengths:
- Cites relevant and up-to-date literature, providing a strong foundation for the study.
Critical Issues:
- Some references seem heavily weighted towards Chinese medicine or related fields, potentially neglecting broader perspectives.
Overall Recommendations
- Clarity and Conciseness: Simplify overly technical sections and reduce redundancy across the manuscript, particularly in the introduction and results sections.
- Accessibility: Ensure that technical methods and results are presented in a way that is accessible to readers from multidisciplinary backgrounds.
- Focus on Novelty: Explicitly highlight what makes this study unique compared to existing research.
- Enhanced Discussion: Provide more depth on the implications for clinical applications and drug development.
- Figures and Tables: Simplify visualizations and provide clear legends to improve readability.
The English could be improved to more clearly express the research.
Author Response
Point to Point Response
Many thanks for the kindhearted review and constructive comments from reviewer and editor. We revised the manuscript carefully, and hope we already addressed all the concerns of editor and reviewer this time. Please see our point-to-point response which shown in below.
Reviewer 3
Comment 1: The manuscript titled “The multi-target action mechanism for the anti-periodontitis effect of Astragali radix based on bioinformatics analysis and in vitro verification” explores the therapeutic potential of Astragali radix in periodontitis treatment. The study highlights its anti-inflammatory and anti-apoptotic mechanisms, focusing on Kaempferol as a key compound and identifying seven core targets (e.g., IL6, TNF, CASP3). Molecular docking and in vitro experiments confirm its ability to inhibit LPS-induced apoptosis and reduce inflammatory marker expression. These findings provide a promising basis for the targeted use of Astragali radix in periodontitis.
Comment 2: Several critical aspects need attention to improve its clarity, coherence, and overall impact.
Unified Response: Thanks for your comments.
Strengths:
Introduction
Comment 1: Provides a solid background on periodontitis and the relevance of Astragali radix in Traditional Chinese Medicine (TCM).
Comment 2: Justifies the study by highlighting gaps in current research on the molecular mechanisms of Astragali radix.
Methods and Materials
Comment 1: Thorough explanation of methodologies such as network pharmacology, molecular docking, and in vitro experiments.
Comment 2: Comprehensive descriptions of software and databases used.
Results
Comment 1: Systematic presentation of findings with appropriate use of tables, figures, and networks.
Comment 2: The progression from identifying active ingredients to validating their molecular effects is logical.
Discussion
Comment 1: Adequate integration of results with previous literature to support conclusions. Comment 2: Provides insights into the therapeutic potential of Kaempferol and other active compounds.
Conclusion
Comment 1: Summarizes the main findings effectively.
Comment 2: Reiterates the potential of Astragali radix as a therapeutic agent for periodontitis.
References
Comment 1: Cites relevant and up-to-date literature, providing a strong foundation for the study.
Unified Response: We sincerely appreciate your recognition of our research efforts.
Critical Issues:
Introduction
Comment 1: Redundant explanations about TCM and periodontitis could be trimmed.
Answer 1: Thank you for your comments. Based on your suggestions, we have removed redundant sections regarding TCM and Periodontitis from the introduction.
Comment 2: The link between bioinformatics approaches and traditional medicine could be better articulated.
Answer 2: We sincerely appreciate your valuable suggestions. The complexity of components and unclear mechanisms of traditional Chinese medicine have limited its further drug development. Bioinformatics employs computational techniques to extract data from public online databases, which can be used to analyze the effects of multi-component drugs on the human system. This aids in identifying therapeutic targets of effective drug components, enhancing drug efficacy, reducing side effects, and shortening the research timeline. Relevant content has been appropriately revised in the text, with modifications labelled in blue.
Comment 3: The novelty of the research is implied but not explicitly stated.
Answer 3: We sincerely appreciate your insightful suggestions. Research suggests that Astragali radix possesses the potential to treat periodontitis; however, its composition is intricate, and the specific mechanisms remain inadequately elucidated. The novelty of this study resides in the integration of bioinformatics and in vitro experiments, which aim to preliminarily validate the primary mechanisms through which Astragali radix may combat periodontitis. The relevant content has been revised in the manuscript, with the modified sections labelled in blue for clarity.
Methods and Materials
Comment 1: Some sections, such as molecular docking and dynamics simulations, are overly technical and may not be accessible to a broader audience.
Answer 1: We sincerely appreciate the valuable suggestions you provided. Following your suggestion, we referenced relevant literature (Ferreira, L.G., Molecules 2015), and revised the description of molecular docking and molecular dynamics simulations in the Materials and Methods section, to enhance its readability and comprehensibility for a broad audience. The modifications have been labelled in blue.
Comment 2: Statistical methods need further clarification, particularly how key thresholds (e.g., p-values) were chosen.
Answer 2: We appreciate your valuable suggestions. According to the literature (Gong, W, Nutrients 2024, 16) and standard statistical methods, a P-value < 0.05 is considered indicative of statistical significance in differential gene expression, suggesting that the identified differential genes are associated with distinct biological processes and signaling pathways. Additionally, in the differential gene selection process, the range of differential genes is refined by adjusting the threshold of |logFC| while maintaining a P-value < 0.05.
Comment 3: No mention of reproducibility or potential limitations of the computational models.
Answer 3: We appreciate your valuable suggestions. In this study, the computational models used for molecular docking (Cheng, T, J Chem Inf Model 2009) and molecular dynamics (Xia, Q.D., Cell Prolif 2020) have been widely applied in previous research and are relatively mature. Although these computational models still have some shortcomings, they are still widely used. It is hoped that in the near future, with the collective efforts of a large number of scientists and engineers, these computational models can be further improved.
Results
Comment 1: Certain figures and tables (e.g., heatmaps, docking results) are overly complex and lack concise legends for easier interpretation.
Answer 1: We sincerely appreciate the valuable suggestions you have provided. We have conducted a thorough and comprehensive review of the legends for each figure. Furthermore, we have included concise descriptions to enhance readability and comprehension for a broader audience. All modifications are labelled in blue.
Comment 2: Overemphasis on numerical values (e.g., docking scores, p-values) without sufficient biological interpretation.
Answer 2: We appreciate your comments. During the molecular docking process, small molecules representing drug active ingredients are docked with larger molecules that serve as disease targets. This simulation allows for the analysis of the binding forces involved and the subsequent scoring of the docking results. A higher molecular docking score suggests that the complex formed by the interaction of small active ingredient molecules with larger disease target molecules is more stable, thereby enhancing the therapeutic effect. In the analysis of differentially expressed genes, as well as Gene Ontology and Kyoto Encyclopedia of Genes and Genomes enrichment, a P value < 0.05 is deemed statistically significant. This indicates that the selected differentially expressed genes are associated with biological processes and signaling pathways that exhibit significant statistical differences. Therefore, we conclude that the differentially expressed genes and signaling pathways identified with P < 0.05 represent potential biological processes involved in disease development. Targeting these differential genes or signaling pathways may offer valuable insights for disease treatment.
Comment 3: The narrative often shifts between technical details and broader implications without clear transitions.
Answer 3: We appreciate your valuable suggestions. In response to your suggestions, we have revised the description of the results section and the associated figures to facilitate a more natural and seamless transition between the technical details and their broader biological significance. The modified sections have been labelled in blue for clarity.
Discussion
Comment 1: The discussion is lengthy and sometimes reiterates results rather than interpreting them.
Answer 1: We greatly appreciate your insightful comments. In response to your suggestions, we have refined the discussion section to concentrate exclusively on the key research findings. All modifications have been labelled in blue for clarity and ease of reference.
Comment 2: Potential clinical applications are mentioned but not deeply explored.
Answer 2: We appreciate your valuable suggestions. Through bioinformatics analysis and in vitro experimental results, we identified that the primary active component of Astragali radix, formononetin, targets CASP3, IL-6, and MMP9, thereby exerting its anti-periodontitis effects. Building upon these findings, we aim to conduct further research, including the incorporation of formononetin into hydrogels, 3D-printed scaffolds, and electro-spun matrices, to develop drug delivery systems and evaluate their local or systemic applications. Our research team is currently focused on these areas and anticipates reporting relevant experimental results in subsequent studies. Nevertheless, we continue to encounter numerous challenges in the associated preclinical experiments.
Comment 3: Limitations of the study are acknowledged but could be expanded to include the generalizability of in vitro findings.
Answer 3: We appreciate your valuable suggestions. The in vitro experimental section of this study primarily employed standard techniques, including stem cell identification, cell viability assessment, and protein immunoblotting. However, there remains a significant gap in relevant research concerning in vitro gene knockout cell models and protein-protein interaction models. Future experiments should focus on conducting further research to elucidate the anti-periodontitis effects of kaempferol from a more comprehensive perspective. The modifications have been labelled for clarity.
Conclusion
Comment 1: The conclusion is overly general and lacks specific actionable recommendations for future research.
Answer 1: We sincerely appreciate the valuable suggestions you provided. Based on preliminary results from bioinformatics analyses and in vitro experiments, we draw the following conclusions: Firstly, Astragali radix exerts anti-periodontitis effects through various components, including kaempferol and quercetin, which act on multiple targets associated with periodontitis, such as IL-6 and MMP-9. Secondly, additional in vitro experiments validated that kaempferol significantly reduced the expression levels of inflammatory factors, including IL-6, CASP3, and MMP-9, in LPS-induced human periodontal ligament stem cells. These research findings provide valuable guidance for the further development of local or systemic formulations of kaempferol, utilizing methods such as hydrogels, electrospinning, or 3D printing of scaffolds. The modifications have been labelled in blue within the manuscript.
Comment 2: The statement on “scientific evidence for related drug development” could benefit from clarification or examples.
Answer 2: We appreciate your valuable suggestions. Based on our research findings, one of the active components of Astragali radix, specifically kaempferol, demonstrates anti-periodontal effects. However, given that kaempferol is a flavonoid compound characterized by low oral bioavailability and low local drug concentration following systemic administration, it is crucial to develop appropriate carriers for the formulation of localized medications. This indicates that it may provide a scientific foundation for informing drug development.
References
Comment 1: Some references seem heavily weighted towards Chinese medicine or related fields, potentially neglecting broader perspectives.
Answer 1: We sincerely appreciate your insightful feedback. Following a thorough review of the relevant literature, we have incorporated additional references pertaining to in vitro experiments and bioinformatics, etc. The newly added reference numbers include: [13], [26], [39], [40], [41] etc. The modified sections have been distinctly labelled in blue.
Overall Recommendations
Comment 1: Clarity and Conciseness: Simplify overly technical sections and reduce redundancy across the manuscript, particularly in the introduction and results sections.
Answer 1: We greatly appreciate the insightful suggestions you provided. In response to your suggestions, we have revised the pertinent sections of the manuscript. The revised sections have been labelled in blue for your convenience.
Comment 2: Accessibility: Ensure that technical methods and results are presented in a way that is accessible to readers from multidisciplinary backgrounds.
Answer 2: We sincerely appreciate the valuable suggestions you provided. We have revised certain technical methods in the text to be described in more accessible language, allowing a broader range of readers to understand them more easily. The revised sections are labelled in blue.
Comment 3: Focus on Novelty: Explicitly highlight what makes this study unique compared to existing research.
Answer 3: We sincerely appreciate your valuable suggestions. We have elaborated on the novelty of this research in the relevant sections of the manuscript. The modifications have been highlighted in blue for clarity.
Comment 4: Enhanced Discussion: Provide more depth on the implications for clinical applications and drug development.
Answer 4: We sincerely appreciate your insightful suggestions. In response to your suggestions, we have revised the relevant sections of the manuscript. The modifications have been highlighted in blue for your convenience.
Comment 5: Figures and Tables: Simplify visualizations and provide clear legends to improve readability.
Answer 6: We sincerely appreciate your insightful suggestions. We have revised the legends in the manuscript, and the modifications have been labelled in blue.
Comments on the Quality of English Language
Comment 1: The English could be improved to more clearly express the research.
Answer 1: We sincerely appreciate the valuable suggestions you provided. The author of this paper, Professor Cory J. Xian from the Unisa Clinical and Health Sciences at the University of South Australia, has revised the manuscript to improve its linguistic quality. All modifications have been labelled in blue for clarity.
